# The Importance of Nutraceuticals in COVID-19: What’s the Role of Resveratrol?

**DOI:** 10.3390/molecules27082376

**Published:** 2022-04-07

**Authors:** Elisa Domi, Malvina Hoxha, Entela Kolovani, Domenico Tricarico, Bruno Zappacosta

**Affiliations:** 1Department for Chemical-Toxicological and Pharmacological Evaluation of Drugs, Faculty of Pharmacy, Catholic University Our Lady of Good Counsel, Rruga Dritan Hoxha, 1000 Tirana, Albania; e.domi@unizkm.al (E.D.); m.hoxha@unizkm.al (M.H.); 2Infectious Diseases Department, Faculty of Medicine, University of Medicine, Tirana, Rruga e Dibrës, 1005 Tirana, Albania; vlasiovi@hotmail.com; 3Section of Pharmacology, Department of Pharmacy-Pharmaceutical Sciences, University of Bari, Via Orabona 4, 70125 Bari, Italy; domenico.tricarico@uniba.it

**Keywords:** COVID-19, resveratrol, immune system, inflammation

## Abstract

Since COVID-19 has affected global public health, there has been an urgency to find a solution to limit both the number of infections, and the aggressiveness of the disease once infected. The main characteristic of this infection is represented by a strong alteration of the immune system which, day by day, increases the risk of mortality, and can lead to a multiorgan dysfunction. Because nutritional profile can influence patient’s immunity, we focus our interest on resveratrol, a polyphenolic compound known for its immunomodulating and anti-inflammatory properties. We reviewed all the information concerning the different roles of resveratrol in COVID-19 pathophysiology using PubMed and Scopus as the main databases. Interestingly, we find out that resveratrol may exert its role through different mechanisms. In fact, it has antiviral activity inhibiting virus entrance in cells and viral replication. Resveratrol also improves autophagy and decreases pro-inflammatory agents expression acting as an anti-inflammatory agent. It regulates immune cell response and pro-inflammatory cytokines and prevents the onset of thrombotic events that usually occur in COVID-19 patients. Since resveratrol acts through different mechanisms, the effect could be enhanced, making a totally natural agent particularly effective as an adjuvant in anti COVID-19 therapy.

## 1. Introduction

The current coronavirus COVID-19 or severe acute respiratory syndrome coronavirus 2 (SARS-CoV-2), was first reported in late 2019 in Wuhan (China) and has spread worldwide, rapidly becoming a pandemic [1]. To date, it represents the biggest health problem all over the world, so the development of an effective therapy is urgently required [2]. COVID-19 may be asymptomatic or manifested in three clinical phases, respectively:the initial upper respiratory tract infection that may evolve to a pneumonic phase followed by the hyperinflammatory phase, which may be lethal [3].

One of the main characteristics common to all COVID-19 patients is the complex immune dysregulation resulting in hyperinflammation and hypercytokinemia, otherwise known as cytokine storm [4]. This condition protracted over several days leads to organ injury, followed by organ failure and increased risk of mortality [5]. Severe disease is characterized by a marked inflammation resulting from high circulating cytokines, such as interleukin-6 (IL-6), and tumor necrosis factor (TNF). These inflammatory mediators are responsible for the damaging effects on the immune, hematologic, respiratory, renal, gastrointestinal, and other body systems [6].

The risk factors that increase mortality rate include older age, comorbidity (in particular diabetes, hypertension and cardiac disease), non-asthmatic respiratory disease, obesity, immunosuppression, and male sex [7]. This seems to be due to a deficiency of nicotinamide adenine dinucleotide (NAD^+^). NAD^+^ is the main regulator factor of the silent information regulator of transcription1 (SIRT1); in fact, when the tissue levels of NAD^+^ are increased, also SIRT1 activity is improved [8]. SIRT1 belongs to sirtuins, a family of seven NAD^+^-dependent class III histone deacytalase and mono-ADP-ribosyl transferase [7]. The first function attributed to SIRT1 is the deacetylation of histone proteins. The mechanism of acetylation/deacetylation is involved in the modification and regulation of the activity of proteins, leading SIRT1 to have different functions in different pathologies and biological processes as cell survival, inflammation, oxidative stress, and mitochondrial biogenesis [9].

Therefore, the interventions aimed at increasing its activity and gene expression are of considerable interest. In reference to that, resveratrol (RES) is the prototype of the SIRT1 activators [10].

RES (3,5,4′-trihyroxystilbene) is a polyphenolic compound initially isolated from the roots of *Veratrum grandiflorum* and later from the roots of *Polygonum cuspidatum* [11]. It has a stilbenoid structure and exists in two isomeric forms (*cis* and *trans*), but it is the *trans* form, due to its major stability, that is responsible for the beneficial effects of RES [12]. It is produced by 70 different plant species in response to infection or under stress conditions [12]. The main food sources of RES are red grapes, cranberry, blueberry, peanuts, soy, and wine [4]. In particular, red grapes have an RES content equal to 0.15 mg/100 g, cranberries 3.0 mg/100 g, blueberries 0.67 mg/100 g, peanuts 0.04 mg/100 g and wine 0.67 mg/100 mL [11]. Regarding its physicochemical properties, RES is poorly soluble in water, and its stability at body temperature depends on pH, in fact for pH over 7.4 its stability and solubility exponentially decrease. Once orally administered, about 70% is absorbed in the gastro-intestinal tract, but its bioavailability is low due to its metabolism. In fact RES undergoes glucuronidation in the liver and intestine (trans-resveratrol-3-O-glucuronide), and sulfation (trans-resveratrol-3-O-sulfate) in the liver. In the bloodstream, we find RES in three forms: glucuronide, sulfate, or unmetabolized. In the end, RES forms some complexes limiting its bioavailability, by binding with albumin and lipoprotein [13]. Among the approaches to overcome the problem of reduced bioavailability, there is that of using resveratrol derivatives. They can include newly synthesized derivatives or testing naturally occurring derivatives, which sometimes are normally present in the human diet [14]. The most common choice has been the methylation of the fenolic groups, obtaining for example: 3,4,5,4′-tetramethoxystilbene that showed particularly high accumulation in the intestinal mucosa; Pterostilbene, a dimethyl derivative of RES, with improved bioavailability; also a naturally occurring active analog of resveratrol, 3,5,4′-trimethoxytrans-stilbene with greater plasma exposure, longer half-life, and lower clearance than resveratrol in rats [14].

To date, the only reported side effect was the onset of diarrhea after daily administration of 200 mg of RES, which was not associated with clinical complications [15]. Moreover, the intake of a few milligrams per day does not affect the onset of systemic or intestinal RES-drug interaction, so the use of low-milligram doses of RES as a food supplement should not lead to critical interactions with the intestinal metabolism of co-administered drugs [16]. Things change if we talk about higher doses, with a range of 1 g/day or more. In this case RES can have a marked interaction with drugs that have a large first pass metabolism in the intestine, such as for example some calcium-channel blockers, sildenafil, midazolam and nefazodone. In these patients, the risk of drug interaction may exceed the beneficial effect of RES; therefore, it is preferable to avoid its use [16].

RES is known for its beneficial properties; it has antioxidant, anti-inflammatory, antiviral, anti-aging and life-prolonging effects [17]. It also has immunomodulating properties by interfering with cell regulation and pro-inflammatory cytokine synthesis [18]. Among the different mechanisms involved, RES exerts its activity through SIRT1 [19].

Several studies, regarding the dietary intake of particular nutrients that can increase the immune defenses, are gaining more and more interest, due to the fact that the nutritional profile can influence the patient’s immunity [1].

Indeed, exploring the repurposing of natural compounds may provide alternatives against COVID-19. In fact, different nutraceuticals have a proven ability of immune-boosting, antiviral, antioxidant, and anti-inflammatory effects [1].

As SARS-CoV-2 viral infection is a current global emergency problem, the urgency of finding effective therapies is of great importance. Among the different therapeutic approaches, the use of natural adjuvants has attracted interest, and in particular we have placed our interest in RES, due to its strong anti-inflammatory properties. Based on this scientific evidence, we aimed to collect and review all the information concerning the different roles that RES seems to have in improving the pathophysiology of COVID-19, targeting different pathways.

## 2. Results

### 2.1. SIRT1 and COVID-19

The SARS-CoV-2 hyperinflammatory response is associated with high mortality. SIRT1 among the different roles in the inflammatory response is also a primary defense against DNA and RNA viral pathogens through autophagy mechanisms, which consists in the destruction of damaged cellular components occurring in vacuoles within the cell [20]. SIRT1 also inhibits apoptosis, and protects against hypoxia [20,21,22,23]. The upregulation of SIRT1 decreases viral replication and inhibits the activation of ADAM17, (A Disintegrin and Metalloproteinase Domain 17), also called TNF-α converting enzyme (TACE), by increasing the expression of TIMP3, the gene that encodes for tissue metalloproteinase inhibitor 3 [24].

If ADAM17 expression is not downregulated by SIRT1, TNF-α and IL-6 are released, resulting in an uncontrolled hyperinflammatory response as may happen with COVID-19 [7,24]. SIRT1, by inhibiting ADAM17 and consequently TNF-α and IL-6, exerts an anti-inflammatory action [21,22]. If oxidative stress is severe, increased ADAM17 tries to ameliorate tissue injury by converting active iron (Fe^2+^) to its inert form (Fe^3+^) which is stored in hepatocytes and macrophages. This also potentially transforms haemoglobin to methaemoglobin, reducing its capacity to bind to oxygen [23].

In contrast, a SIRT1 downregulation is associated with increased viral replication, due to enhanced ADAM17 activity causing increased TNF-α, IL-6 and IL-1β levels. Generally increased TNF-α levels increase also SIRT1 activity, but in case of a deficiency of NAD^+^ this would not occur due to an insufficient activation of SIRT1, causing an uncontrolled increase of TNF-α level. SIRT1 maintains also the vascular endothelial function by inhibiting the oxidative stress in vascular endothelial cells (EC), preventing or reducing the potential for the metabolic syndrome, ischaemia–reperfusion injury and inflammation [25]. The endothelial glycocalyx (EG) is a web of membrane-bound glycoproteins on the luminal side of EC that cover the vascular endothelium and exerts anticoagulant properties [26]. The EG separates cellular blood components from the endothelium and maintains the osmotic tension of the intravascular compartment [25]. SARS-CoV-2, as a pathological condition characterized by a hyperinflammatory status, may cause damage to the EG, exposing the endothelium, consequently allowing the adhesion and activation of platelets with degranulation and release of vasoactive molecules [26].

Another mechanism underlying the beneficial effect of RES as a SIRT1 activator involves NRF2 protein. In fact, the activation of SIRT1 also leads to NRF2 up-regulation since it is deacetylated by Sirtuins [27]. NRF2 is an important protein with marked anti-inflammatory and antioxidant properties that may protect the cell from virus consequences. Thus, the use of SIRT1 activators, such as RES, or directly of NRF2 leads to an improvement in the pathophysiology of COVID-19 [27].

NAD^+^ levels decrease with age and are also reduced in conditions associated with oxidative stress as in hypertension, diabetes, and obesity. The same groups were found to be more vulnerable against COVID-19. Low NAD^+^ levels in different tissues lead to cytokine storm due to a decreased activation of SIRT1 [28,29]. Since the EG is already compromised in systemic inflammatory status, such as diabetes, hyperglycemia, surgery, and trauma, under conditions of more severe oxidative stress, as in hyperinflammation COVID-19 mediated, the damage may lead to its destruction and consequently to capillary leak, platelet aggregation, hypercoaguability, and a loss of vascular responsiveness [30].

Lymphopenia is another marker of COVID-19 [31]. The transcriptional profiling of peripheral blood mononuclear (PBMC) cells from SARS-CoV-2-infected patients show an increase in p53 signaling pathway, suggesting the involvement of cell apoptosis in the pathogenesis of COVID-19 [32]. In this regard, Bordoni et al., analyzed p53 and SIRT1 levels in patients with COVID-19 and in the healthy ones [33]. PBMC in COVID-19 patients showed significantly higher p53 expression compared to healthy patients. Conversely, the expression of the deacetylase SIRT1 was decreased in COVID-19 patients, and was negatively correlated with p53 [33]. These data suggest that COVID-19 can be characterized by an increase in p53 transcript in circulating lymphocytes, and by a permanent activated p53 form, probably due to the low level of SIRT1 [33].

Overall, these observations demonstrate that SIRT1 may be a crucial factor in the prevention of the hyperinflammatory response, and may be necessary for a successful defense against viral attack.

### 2.2. Resveratrol and COVID-19

Resveratrol is largely known for several properties and, in particular, in this context, we elucidated its role in SARS-CoV-2 infection. The antiviral activity of RES could have different targets including inflammation, virus-induced apoptosis [2] and autophagy [6,34]; immune system stimulation accompanied by an increase of natural killer immune cells [2]; regulation of renin-angiotensin system (RAS) and expression of angiotensin-converting enzyme 2 (ACE2) which could reduce the virus entrance in the cell [2,35,36]; anti-inflammatory effect sirtuin 1 (SIRT1)-mediated due to the reduction of cytokines released by the viral infection and antioxidant activity thanks to its scavenger role against oxygen free radicals (ROS) [2,37,38].

Besides respiratory problems, a characteristic feature of COVID-19 infection is represented by the alteration of vascular homeostasis accompanied by coagulation disorders, endotheliopathy and inflammation [39]. Thanks to its antithrombotic, anticoagulant and anti-inflammatory effects, RES may mitigate this condition slowing the progress of the disease [39]. However, RES has a poor oral bioavailability and this makes it difficult to use in clinical practice. In fact, after oral administration, resveratrol is rapidly metabolized, thus lowering its bioavailability [13,40,41,42]. To overcome these limitations, several methods have been studied. Amongst them, especially for the respiratory tract, aerosolized suspensions like a resveratrol-containing spray and co-spray dried microparticles may show beneficial effects [42]. RES has been combined with carboxy methylated (1,3/1,6)-β-D glucan (CM-glucan) in a nasal spray which transports RES through the respiratory tract. Specifically, the active medication was represented by an isotonic solution containing resveratrol 0.05% (extracted by *Polygonumcuspidatum*) and CM-glucan 0.33%. [41]. CM-glucan has a dual action: it lowers the mass median aerodynamic diameter of the particles (2.83 μm versus 3.28 μm of RES alone and 2.96 μm of CM-glucan alone) and enhances RES activity since it stimulates the immune system [40]. The effectiveness of the above-mention nasal spray has been tested in clinical practice in allergic children. Patients were instructed to spray two sprays (100 mL/spray) in the nostril three times/day for 2 months, and it has been observed that RES combined with CM-glucan reduces the severity and relapse of upper respiratory tract infections (or symptoms like itching, sneezing, rhinorrhea, nasal obstruction), which are often virus-related [42]. It has been observed that the mechanism underlying the antiviral effect of this nasal spray involved TLR2 up-regulation. Thanks to its positive effect and safety use, this formulation may be suitable for further study for the use of this nasal spray in the clinical practice of the early stages of COVID-19 patients [40].

Below in Figure 1 there is a summary of the positive effects of RES against COVID-19.

#### 2.2.1. Antiviral Activity of Resveratrol

RES seems to be a potent natural antiviral compound against different types of DNA and RNA viruses [43]. It exerts its antiviral effect by acting through different mechanisms; in fact, RES can inhibit virus entrance in cells [35], viral replication, viral protein synthesis, gene expression, and nucleic acid synthesis [43,44,45].

##### Resveratrol and Virus Entrance in Host Cells

Regarding the entry of SARS-CoV-2 into the host cell, the mechanism depends on ACE2. The viral transmembrane spike glycoprotein (S-protein) binds to ACE2 enzyme expressed in the membrane of the target cell allowing the fusion of the virus membrane with consequent entry and replication of SARS-CoV-2 in the host cell [46]. In support of this theory, Horne and Vohl demonstrated that ACE2 knockout mice were resistant to the viral infection by SARS-CoV-2 [47]. RES acts at this level and disrupts the spike protein—ACE2 complex, inhibiting the cell entrance of the virus. Moreover, the viral infection downregulates the ACE2 receptor, and provides the conversion of angiotensin II (AngII) to angiotensin (Ang) leading to an increase of Ang II levels [48]. ACE2 enzyme is highly expressed in heart, lungs, liver and kidneys, where the increased levels of Ang II lead to a state of hyper-inflammation and organ damage that overall increase the chance of mortality from COVID-19 [49]. Once again, RES, through SIRT1 activation, exerts its beneficial effects by improving ACE2 upregulation, and consequently Ang II reduction [47,50].

##### Resveratrol and Viral Replication

Several studies have highlighted the potential role of RES in the inhibition of viral replication, and have shown that different mechanisms are involved [45]. For example, initially it was observed that RES inhibits in vitro replication of Middle East respiratory syndrome-related coronavirus (MERS-CoV) and influenza virus [45]. The mechanisms proposed involved the ability of RES to decrease the expression of the nucleocapsid, an essential protein for viral replication [34], and its role in the inhibition of the p38 mitogen-activated protein kinases(p38MAPK) and class I phosphatidylinositol 3-kinase (PI3K)/AKT/mammalian target of rapamycin complex 1 (mTORC1) (PI3K/AKT/mTOR) pathways, which takes part in autophagy, apoptosis and inflammation processes [34,45,51,52,53].

Moreover, it emerged that RES improves intracellular zinc entrance in cells [54] and, because increased intracellular zinc levels inhibit viral RNA polymerase, this results in the inhibition of SARS-CoV-2 replication [55]. Starting from this evidence, different experiments have been conducted to verify the efficacy of RES in viral infection by SARS-CoV-2.

In 2021, Yang et al., studied the effect of RES in replication inhibition in Vero cells previously infected with SARS-CoV-2 [56]. Different concentrations of RES were tested, and viral replication was measured 48 h after infection, showing a marked inhibition of SARS-CoV-2 replication with an EC_50_ (half-maximal effective concentration) of 4.48 μM [56].

Pasquerau et al., also tested the in vitro effect of RES in SARS-CoV-2 replication and in human coronavirus (HCoV)-229E, another coronavirus family member [57]. Interestingly, RES significantly decreases the viral replication under non-cytotoxic doses up to 25 µM [57].

Despite the existence of such scientific evidence, further studies are needed to elucidate the mechanisms involved.

#### 2.2.2. Anti-Inflammatory Activity of Resveratrol

Systemic hyper-inflammation is one of the main characteristics of COVID-19 viral infection. In particular, the viral nucleocapsid protein (N) activates the cyclooxygenase 2 (COX-2) enzyme, whereas the envelope protein (E) improves the synthesis of some pro-inflammatory cytokines, such as TNF, IL-1 and IL-6 [58,59]. These conditions caused a marked increase of inflammatory mediators and are responsible for the severity of the disease [60,61].

RES, acting through different mechanisms and mediators, exerts its anti-inflammatory properties ameliorating the pathophysiology of SARS-CoV-2 infection. In fact, it seems to inhibit IL-6, IL-1β and IL-8 interleukins expression, mitigating the inflammatory state [62,63,64]. RES also interacts with nuclear factor erythroid 2-related factor 2 (Nrf2), a transcription factor constitutively expressed in the cytoplasm of cells usually associated with Keap-1, a repressor protein [65]. Under oxidative stress and inflammation conditions, Keap-1 expression decreases, leading to Nrf2 activation. RES works by enhancing Nrf2 activation by reducing Keap-1 expression and activating SIRT1 deacetylase [66]. Once Nrf2 is activated, it dissociates from the cytoplasmatic complex with Keap1 and moves into the nucleus; here, it promotes the transcription of some target genes with antioxidant response element (ARE) which protect cells from oxidative stress and inflammation [67,68].

Another mechanism that explains the anti-inflammatory effect of RES involves the ability of this natural compound to induce autophagy [38]. In fact, the viral particles in the infected cells have been found inside autophagy-like vacuoles which, however, are unable to fuse with the lysosomes, thus altering the virus clearance itself. This condition, together with the succession of pro-inflammatory reactions, leads to further impairment of autophagy [38]. The molecular steps of the autophagy pathway involved the PI3K/AKT/mTORC1, and 5′ AMP-activated Protein Kinase (AMPK) pathways, where mTORC1 inhibits while AMPK promotes autophagy initiation [38]. In contrast, the NAD-dependent deacetylase SIRT1, and the transcription factors fork head box O3 (FOXO3) promote some steps of the autophagy process, from phagophore biogenesis up to the fusion of autophagosomes with lysosomes. SARS-CoV-2 virus inhibits AMPK, reducing the fusion of autophagosomes with lysosomes and it also downregulates mTOR, which promotes the formation of autophagy-like vesicles where the virus replicates [10].

RES exerts its function, inducing autophagy and promoting autophagy flux, through AMPK/SIRT1 activation or PI3K/AKT/mTORC1/2 inhibition [10]. These mechanisms were observed since RES treatment increases the clearance of autophagy–lysosomal substrates and the effect is canceled by the use of some autophagy/lysosome inhibitors [38].

#### 2.2.3. Immunomodulating Activity of Resveratrol

Patients affected by COVID-19 show a marked immune dysregulation characterized by lymphopenia, neutrophilia and increased levels of cytokines, also known as cytokine storm condition [69]. These mediators migrate from the blood vessels to the pulmonary alveoli triggering lung damage accompanied by acute respiratory distress syndrome (ARDS) [70].

Different proinflammatory cytokines are involved such as IL-1β, IL-1RA, IL-7, IL-9, IL-10, FGF, G-CSF, GM-CSF, PDGF, VEGF, IFNγ, TNF and chemokines CXCL8, IP10, MCP1, MIP1α, MIP1β [4] and their increased levels lead to the accumulation of cells and fluid in the respiratory system [71]. Once SARS-CoV-2 patients become infected, the innate and the adaptive immune response are activated, and both cytokines and chemokines are necessary for the maintenance of anti-viral immunity [72,73]. In particular, initially, the virus enters the host cells and it is recognized by pathogen recognition receptors (PRRs) which mediate an increased interferon production. These events lead to the activation of the innate immune response which ultimately results in the increased expression of cytokines and chemokines (especially IL-1Rα, IL1β, IL-6 and TNF) [4].

These chemokines stimulate some innate immune response cells like natural killer cells, dendritic cells, polymorphonuclear leukocytes and monocytes which on the other hand produce other chemokines such as MIG, IP10 and MCP-1 that are necessary for T-lymphocytes recruitment. At this point, the specific adaptive immune response starts: CD4 T-cells send signals to B-cells to initiate the production of specific IgG, IgM and IgA antibodies and to CD8 cytotoxic T-cells to eliminate SARS-CoV-2 [74,75,76,77].

By regulating immune cell response and pro-inflammatory cytokine production, RES seems to be a safe immunomodulatory agent in SARS-CoV-2 viral infection [4]. In fact, it interferes with a large number of mediators irreversibly inhibiting IFNγ and IL-2 by splenic lymphocytes [78]; inhibiting GM-CSF and CXCL8 chemokines release [79]; reducing IL-6, TNFα secretion [80]; downregulating NF-κB in the inflammatory cells of lungs [81]; decreasing cytokines IL-1β, IL-8, IL-12 [82]. Thus, overall, RES calms the cytokine storm and relieves the hyper-inflammation state.

#### 2.2.4. Resveratrol and Its Role in Hemostatic Disorders COVID-19 Related

COVID-19 infection is frequently complicated by relevant thrombotic events due to an altered balance between coagulation activation and fibrinolysis inhibition [83] and to a general state of hyperinflammation leading to endotheliopathy and coagulopathy [84,85]. Among the therapies used to overcome the problem there is the use of anticoagulants which, however, often leads to hemorrhagic events [86,87]. For this reason, the interest in finding natural compounds with anti-inflammatory and anti-thrombotic action has grown, and has shown to interfere both with platelet aggregation, and with coagulation cascade factors [88,89,90].

##### Resveratrol Effects in Platelets Aggregation

Regarding its role as an antiplatelet agent, different mechanisms are involved and one of them is linked to the ability of RES to inhibit the cyclooxygenase 1 (COX-1) enzyme and consequently, also to the production of thromboxane (TXA_2_), a mediator with marked pro-thrombotic and pro-aggregating properties [91].

Another mechanism underlying the antiplatelet effect of RES is correlated to the inhibition of the flow of calcium (Ca^2+^), an important promoter of platelet aggregation rate [92].

Moreover, the antiplatelet effect of RES is also due to its role in enhancing the expression and the activity of the endothelial nitric oxide synthase (eNOS) enzyme. eNOS is responsible for the synthesis of nitrogen monoxide (NO), a gasotransmitter with a crucial role in maintaining vascular homeostasis through its vasodilatatory and antiplateletproperties [93]. However, it is hypothesized that the effect of RES on NO may also be related to the interaction with additional metabolic pathways including SIRT1, AMPK and Nrf2 [93].

##### Resveratrol Effects in Coagulation Cascade

RES also interferes with vascular homeostasis by regulating the coagulation cascade. In particular, in vitro studies in HUVECs (human umbilical vein endothelial cells) demonstrate that it decreases the expression of some thrombosis-associated markers as Von Willebrand factor (vWF), procoagulant factor VIII and P-selectin [94]. COVID-19 patients generally show increased levels of vWF and Factor VIII, which both promote venous thromboembolism (VTE) events [94]. On the other hand, P-selectin facilitates the migration of leukocytes to the damaged site promoting their interaction with platelets [94].

So, the effect of RES in these coagulation factors positively affects coagulopathies that characterize COVID-19 patients (Figure 1).

## 3. Discussion

SARS-CoV-2 infection leads to a series of physiological alterations that can result in multi-organ dysfunction. As COVID-19 is an important global health problem, there is still interest in seeking agents that can mitigate the course of the viral infection. In particular, we focus our interest on RES, a natural compound with multiple beneficial effects [17]. The role of RES in the pathophysiology of SARS-CoV-2 infection concerns its immunomodulatory properties by interfering with cell regulation and pro-inflammatory cytokine synthesis [18]. RES is also a SIRT1 activator. On the other hand, SIRT1 is an enzyme involved in COVID-19 due to its regulator NAD^+^ [7]. NAD^+^ levels decline with age and are also reduced in some conditions such as hypertension, diabetes and obesity, characterized by a strong oxidative stress [7]. Since the activation of SIRT1 is dependent on the availability of NAD^+^, their decreased levels are associated with decreased SIRT1 activity too, hence nutritional support with SIRT1 activators as RES, could minimize disease severity [7].

RES has antiviral activity since it inhibits virus entrance in cells [35], through both ACE-dependent [46], and SIRT1-dependent mechanisms [47]. RES also inhibits viral replication via different pathways: by decreasing viral nucleocapsid expression [45]; by promoting intracellular zinc levels that in turns inhibit viral RNA polymerase [31]; and through the inhibition of PI3K/AKT/mTOR pathways which take part in autophagy, apoptosis, and inflammation processes [34,45,53].

The importance of RES is also due to its role as an anti-inflammatory agent since it promotes autophagy [38] and inhibits the expression of pro-inflammatory agents, such as IL-6, IL-1β and IL-8 interleukins, decreasing the inflammatory process caused bySARS-CoV-2 infection [62,63].

Patients affected by COVID-19 show an important immune dysregulation known as cytokine storm condition [69]. RES regulates immune cell response and pro-inflammatory cytokine interfering with different mediators, inhibiting IFNγ and IL-2 [78]; GM-CSF and CXCL8 chemokines release [79]; decreasing IL-6, TNFα secretion [80]; downregulating NF-κB in the inflammatory cells of lungs [81]; decreasing cytokines IL-1β, IL-8, IL-12 [82].

One of the major complications resulting from the viral infection is the onset of thrombotic events, and once again RES is able to mitigate this condition as it regulates the coagulation cascade [39] and acts as an antiplatelet agent through a COX-1 dependentmechanism [92] and eNOS dependent mechanism [93].

Overall, the various mechanisms involved act synergistically to improve the pathophysiology of viral infection of SARS-CoV-2, suggesting that RES intake might have a beneficial effect.

## 4. Materials and Methods

We searched PubMed and Scopus databases as main sources using different keywords in order to identify all the published data concerning the correlation between COVID-19 and resveratrol and the mechanisms underlying this correlation. The keywords were as follows: “Resveratrol and its natural source”, “Physicochemical properties of resveratrol”, “Pharmacokinetics and metabolism of resveratrol”, “Resveratrol and drug interactions”, “Adverse reactions of resveratrol”, “Resveratrol analogues”,“COVID-19 and resveratrol”, “COVID-19 and polyphenols”, “COVID-19 and nutraceuticals”, “COVID and SIRT1”. Thirty-two articles were eligible and were analyzed in order to summarize the actual information about the role of resveratrol in ameliorating the pathophysiology of COVID-19.

## 5. Conclusions

In summary, this review aimed to clarify the association between RES intake and COVID-19 infection. SARS-CoV-2 infection is characterized by an alteration of immune system and hyperinflammation, which protracted over time leads to an increased risk of mortality, especially in some vulnerable categories of patients with hypertension, diabetes and obesity. This is due to decreased NAD^+^ levels that consequently lead to SIRT1 downregulation. Since SIRT1 positively affects COVID-19, its activators may be a potential adjuvants therapy. RES is the main activator of SIRT1.On the other hand, RES is considered an optimal natural compound known for its antioxidant, anti-inflammatory, antiviral, anti-aging and life-prolonging effects. Since it acts through different mechanisms, the effect could be enhanced, making a totally natural agent particularly effective as an adjuvant in anti COVID-19 therapy. However, the poor oral bioavailability could limit its use; hence, further studies are needed.

## Data Availability

Not applicable.

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
