# Peer review of "The Importance of Nutraceuticals in COVID-19: What’s the Role of Resveratrol?"

_molecules, 2022, doi:10.3390/molecules27082376_

Round 1
Reviewer 1 Report
The manuscript entitled “Nutraceuticals importance in COVID-19: what is the role of resveratrol?” have demonstrated significant review studies of association between resveratrol (RES) intake and COVID-19 infection. The subject of the study is current. It describes the role of resveratrol, a compound that is an important factor in the non-enzymatic system for neutralizing free radicals, in the pathophysiology of Covid-19.
The study was written carefully and well in terms of language. In my opinion manuscript should be accepted.
Author Response
The manuscript entitled “Nutraceuticals importance in COVID-19: what is the role of resveratrol?” have demonstrated significant review studies of association between resveratrol (RES) intake and COVID-19 infection. The subject of the study is current. It describes the role of resveratrol, a compound that is an important factor in the non-enzymatic system for neutralizing free radicals, in the pathophysiology of Covid-19.
The study was written carefully and well in terms of language. In my opinion manuscript should be accepted.
Answer: Thank you very much for your comments.

Reviewer 2 Report
The authors raised a very interesting and vital problem regarding the possible potentials of therapy for covid-19 with resveratrol. Currently, several works have been published on this topic, however, the proposed review differs in that it deeply illuminates the aspect of the effects of resveratrol associated with the activation of SIRT. The review is very good, but it can be improved, in my humble opinion, by additionally highlighting the following point.
- 1) Are there any examples of clinical use or clinical trials of resveratrol or preparations containing it in the treatment of acute covid or long-covid? In other words, say some words on clinical practice with resveratrol (see, for example, G.A. Rossi, O. Sacco, A. Capizzi, P. Mastromarino. Can resveratrol-inhaled formulations be considered potential adjunct treatments for COVID-19? Front. Immunol. 12 (May) (2021) 1–11; Role of Resveratrol in Prevention and Control of Cardiovascular Disorders and Cardiovascular Complications Related to COVID-19 Disease: Mode of Action and Approaches Explored to Increase Its Bioavailability. Molecules 2021, 26 (10). DOI: 10.3390/molecules26102834.)
- 2) Search keywords are reasonable and well-chosen, however, for example, they do not consider articles such as, for example, Khan, H.; Patel, S.; Majumdar, A., Role of NRF2 and Sirtuin activators in COVID-19. Clin Immunol 2021, 233, 108879. DOI: 10.1016/j.clim.2021.108879. It seems to me that this review makes sense to include in references.
- 3) I also advise you to add an excellent review to the list of references Liao, M. T.; Wu, C. C.; Wu, S. V.; Lee, M. C.; Hu, W. C.; Tsai, K. W.; Yang, C. H.; Lu, C. L.; Chiu, S. K.; Lu, K. C., Resveratrol as an Adjunctive Therapy for Excessive Oxidative Stress in Aging COVID-19 Patients. Antioxidants (Basel) 2021, 10 (9). DOI: 10.3390/antiox10091440
Author Response
The authors raised a very interesting and vital problem regarding the possible potentials of therapy for covid-19 with resveratrol. Currently, several works have been published on this topic, however, the proposed review differs in that it deeply illuminates the aspect of the effects of resveratrol associated with the activation of SIRT. The review is very good, but it can be improved, in my humble opinion, by additionally highlighting the following point.
1) Are there any examples of clinical use or clinical trials of resveratrol or preparations containing it in the treatment of acute covid or long-covid? In other words, say some words on clinical practice with resveratrol (see, for example, G.A. Rossi, O. Sacco, A. Capizzi, P. Mastromarino. Can resveratrol-inhaled formulations be considered potential adjunct treatments for COVID-19? Front. Immunol. 12 (May) (2021) 1–11; Role of Resveratrol in Prevention and Control of Cardiovascular Disorders and Cardiovascular Complications Related to COVID-19 Disease: Mode of Action and Approaches Explored to Increase Its Bioavailability. Molecules 2021, 26 (10). DOI: 10.3390/molecules26102834.)
Answer: In the article it has been added an example on the clinical use of resveratrol with improved bioavailability, since oral administration does not lead to sufficient therapeutic concentrations (all changes are evidenced in red).
2) Search keywords are reasonable and well-chosen, however, for example, they do not consider articles such as, for example, Khan, H.; Patel, S.; Majumdar, A., Role of NRF2 and Sirtuin activators in COVID-19. Clin Immunol 2021, 233, 108879. DOI: 10.1016/j.clim.2021.108879. It seems to me that this review makes sense to include in references.
Answer: The suggested reference has been added.
3) I also advise you to add an excellent review to the list of references Liao, M. T.; Wu, C. C.; Wu, S. V.; Lee, M. C.; Hu, W. C.; Tsai, K. W.; Yang, C. H.; Lu, C. L.; Chiu, S. K.; Lu, K. C., Resveratrol as an Adjunctive Therapy for Excessive Oxidative Stress in Aging COVID-19 Patients. Antioxidants (Basel) 2021, 10 (9). DOI: 10.3390/antiox10091440
Answer: The suggested reference has been added.

Reviewer 3 Report
Title
Nutraceuticals importance in COVID-19: what is the role of 2 resveratrol? "what” in the title should be written as “What”
Abstract
In line 10, “Since COVID-19 has affected global public health” should be written as “Since COVID-19 has affected global public health,”
In line 13, “increases the risk of mortality” should be written as “increases the risk of mortality,”
In line 13, “consequently” should be written as “consequently,”
In line 14, “patient’s immunity” should be written as “patient’s immunity,”
In line 17, “Interestingly” should be written as “Interestingly,”
Introduction
In line 30, “To date” should be written as “To date”
In line 135, reference should be included.
In line 145, the sentence should be referenced.
In line 175, the sentence should be referenced.
In line 189, the reference should be re-written.
In line 192, “EC50” should be written as “EC50”
In line 195, “Interestingly” should be written as “Interestingly,”
In line 202, “In particular” should be written as “In particular,”
In line 212, “Under oxidative stress and inflammation conditions” should be written as “Under oxidative stress and inflammation conditions,”
I think it would be most appropriate to use a schematic diagram to depict the mechanism of action of resveratrol as an antiviral agent.
All the below mechanism of actions should be represented in schematic diagram.
in fact it has antiviral activity inhibiting virus entrance in cells and viral replication; resveratrol also improves autophagy and decreases pro-inflammatory agents expression acting as an anti-inflammatory agent; it regulates immune cell response and pro-inflmmatory cytokines and prevents the onset of thrombotic event that usually occur in COVID-19 patients
Author Response
- Title
Nutraceuticals importance in COVID-19: what is the role of 2 resveratrol? "what” in the title should be written as “What”
Answer: The title was modified
- Abstract
Answer: In line 10, “Since COVID-19 has affected global public health” was replaced with “Since COVID-19 has affected global public health,”
Answer: In line 13, “increases the risk of mortality” was replaced with “increases the risk of mortality,”
Answer: In line 13, “consequently” was replaced with “consequently,”
Answer: In line 14, “patient’s immunity” was replaced with “patient’s immunity,”
Answer: In line 17, “Interestingly” was replaced with “Interestingly,”
- Introduction
Introduction
In line 30, “To date” was replaced with “To date,”
In line 135, reference has been included.
In line 145, the sentence should be referenced.
In line 175, the sentence should be referenced.
In line 189, the reference should be re-written.
In line 192, “EC50” was replaced with “ECâ‚…â‚€”
In line 195, “Interestingly” was replaced with “Interestingly,”
In line 202, “In particular” was replaced with“In particular,”
In line 212, “Under oxidative stress and inflammation conditions” was replaced with “Under oxidative stress and inflammation conditions,”
Answer: The respective changes were done
I think it would be most appropriate to use a schematic diagram to depict the mechanism of action of resveratrol as an antiviral agent. All the below mechanism of actions should be represented in schematic diagram. In fact it has antiviral activity inhibiting virus entrance in cells and viral replication; resveratrol also improves autophagy and decreases pro-inflammatory agents expression acting as an anti-inflammatory agent; it regulates immune cell response and pro-inflmmatory cytokines and prevents the onset of thrombotic event that usually occur in COVID-19 patients
Answer: The requested diagram was added.
Reviewer 4 Report
The manuscript authored by Elisa Domi et al., is clear and easy to follow. Explore a topic of current importance. however, it is necessary to improve the methodological keywords:
· natural source of resveratrol
· pharmacokinetics
· oral bioavailability
· metabolism
· drug interactions
· adverse reactions
· resveratrol analogs, pharmaceutical forms available
The holistic approach will allow generating a comprehensive and complete vision of resveratrol as an adjuvant in the treatment of COVID-19 infection.
Review attached file

Author Response
The manuscript authored by Elisa Domi et al., entitled: Nutraceuticals importance in COVID-19: what is the role of resveratrol? in Special Issue: Resveratrol News & Views: From the Molecular Mechanism to Nutritional and Biomedical Applications, is clear and easy to follow. Explore a topic of current importance. The authors have reviewed the bibliography concerning the role of resveratrol as an immunomodulator and anti-inflammatory agent. The analysis of several documents referring to: · Silent information regulator of transcription1 (SIRT1) vs. COVID-19. · Resveratrol vs. COVID-19. · The antiviral activity of resveratrol (entrance, viral replication). · Mechanisms of anti-inflammatory activity. · Mechanisms of immunomodulatory activity (pro-inflammatory cytokines, cell response). · Hemostatic disorders COVID-19. · Effects on platelets aggregation · Effects in the coagulation cascade. Based on the author's findings, resveratrol exerts an antiviral activity and inhibits virus entrance. Similarly, inhibits viral replication, interferes with cell regulation and pro-inflammatory cytokine synthesis. On the other hand, resveratrol is also a SIRT1 activator (SIRT1 is an enzyme involved in COVID-19 due to its regulator NAD+ and NAD+ levels decline with age, hypertension, diabetes, and obesity). In this context, nutritional support with SIRT1 activators (resveratrol) could minimize disease severity. In this context, it is necessary to improve the methodological keywords: Natural source of resveratrol Physicochemical properties of resveratrol Pharmacokinetics Oral bioavailability Metabolism Drug interactions Adverse reactions Resveratrol analogues The holistic approach will allow generating a comprehensive and complete vision of resveratrol as an adjuvant in the treatment of COVID-19 infection. The results section fails in giving numerical data (potency, efficacy) of antiviral, anti-inflammatory, and immunomodulating activity induced by resveratrol. The following table lists observations to the document.
Answer: After adding to the methodological section the suggested keywords regarding resveratrol, we improved its description in the introduction section of the manuscript (all changes are evidenced in red)
- Observation of the table
In line 2 the sentence has been corrected.
In line 35 the sentence has been corrected.
In line 81 the sentence has been corrected.
In line 111 the sentence has been corrected.
In line 120 the sentence has been corrected.
In line 130 the sentence has been corrected.
In line 134 the sentence has been corrected.
In line 155 the sentence has been corrected.
In line 160 the sentence has been corrected.
In line 174 the sentence has been corrected.
In line 198 the sentence has been corrected.
In line 201 the sentence has been corrected.
In line 219 the sentence has been corrected.
In line 239 the sentence has been corrected.
In line 268 the sentence has been corrected.
In line 277 the sentence has been corrected.
In line 282 the sentence has been corrected.
In line 291 the sentence has been corrected.
In line 304, in the figure number 1 a 2D structure of resveratrol has been added.
In line 309 the sentence has been corrected.
In line 315 the sentence has been corrected.
In line 316 the sentence has been corrected.
In line 318 the sentence has been corrected.
In line 345 the sentence has been corrected.
In line 353 the sentence has been corrected.
Answer: We would like to thank the reviewer of the precious comments on improving our manuscript. The respective changes were done.
Round 2
Reviewer 4 Report
- Reinforce descriptions about physicochemical properties of resveratrol. Especially those involved in aqueous solubility.
- Include descriptions about the transport and metabolism of resveratrol in the human respiratory tract.
- Is it feasible to consider the merger of Diagram 1 and Table 1?.
The following table lists observations to the document.

Author Response
- As recommended the description about some physicochemical properties of resveratrol is reinforced.
- The description about the transport and metabolism of resveratrol in the respiratory tract has been provided.
- Since the Diagram involves the information of the Table, we think it is suitable to delete the Table at all.
- Based on the observations of the table provided by the Referee all the sentences have been corrected and the data required has been added.